# Structural insights into MIC2 recognition by MIC2-associated protein in *Toxoplasma gondii*

Su Zhang[1,5], Fangfang Wang [1,5], Dujuan Zhang[1,5], Dongsheng Liu[2], Wei Ding [3], Timothy A. Springer [4] & Gaojie Song [1✉]

Microneme protein 2 (MIC2) and MIC2-associated protein (M2AP) play crucial roles in the gliding motility and host cell invasion of *Toxoplasma gondii*. Complex formation between MIC2 and M2AP is required for maturation and transport from the microneme to the parasite surface. Previous studies showed that M2AP associates with the 6th TSR domain of MIC2 (TSR6), but the detailed interaction remains unclear. In this study, we report crystal structures of M2AP alone and in complex with TSR6. TSR domains have an unusually thin, long structure with a layer of intercalated residues on one side. The non-layered side of TSR6 with hotspot residue His-620 at the center binds to M2AP. Remarkably, we show that TSR6 residue Y602 is dynamic; it equilibrates between being part of the layer (the layered state) and in a flipped-out state in the absence of M2AP. However, when bound to M2AP, Y602 shifts to the flipped-out state. Our findings provide insights into the association and stabilization of MIC2-M2AP complex, and may be used to develop new therapies to prevent infections caused by this parasite.

[1] Shanghai Frontiers Science Center of Genome Editing and Cell Therapy, Shanghai Key Laboratory of Regulatory Biology, Institute of Biomedical Sciences and School of Life Sciences, East China Normal University, 200241 Shanghai, China. [2] iHuman Institute, ShanghaiTech University, 201210 Shanghai, China. [3] Beijing National Laboratory for Condensed Matter Physics, Institute of Physics, Chinese Academy of Sciences, 100190 Beijing, China. [4] Program in Cellular and Molecular Medicine, Boston Children's Hospital, and Departments of Biological Chemistry and Molecular Pharmacology and of Medicine, Harvard Medical School, Boston, MA 02115, USA. [5]These authors contributed equally: Su Zhang, Fangfang Wang, Dujuan Zhang. ✉email: gjsong@bio.ecnu.edu.cn

*T*oxoplasma gondii is a parasite that causes toxoplasmosis, a potentially fatal disease in immuno-compromised individuals[1]. Like other members of the phylum *Apicomplexa*, *Toxoplasma* uses gliding motility to migrate and invade host cells. In *Apicomplexa*, proteins required for gliding and infection are secreted from micronemes located at the parasite's apical end[2]. The primary adhesin orthologues are known as microneme protein 2 (MIC2) in *Toxoplasma* and thrombospondin repeat anonymous protein (TRAP) in *Plasmodium*. MIC2 and TRAP are type I transmembrane proteins that bind to extracellular ligands via ectodomains and connect to the motility apparatus via cytoplasmic domains, enabling both gliding motility and host cell invasion (Fig. 1a)[3–5].

MIC2-associated protein (M2AP) is composed of the secretory leader peptide (SLP), propeptide (Pro), β-domain, and Coil domain (Fig. 1a)[6], which is necessary for MIC2 transport through the secretory network[6–8], and knockout or protein level regulation studies have shown the critical role of the MIC2-M2AP complex[7,8]. MIC2 contains a von Willebrand factor type A (VWA) domain and six tandem thrombospondin type 1 repeat (TSR) domains in its ectodomain. Previous structures of MIC2 fragments containing the VWA-TSR1 domains have revealed an unusually long TSR1 domain that associates with its preceding VWA domain in a potentially closed conformation[9]. The VWA domain plays a crucial role in binding to its ligand on the host cell surface, and its homologous domain in integrins, the inserted I domain has both a high-affinity open conformation and a low affinity closed conformation[10,11]. Both conformations have been defined in the Plasmodium TRAP orthologue, and the conformational change of TRAP has been suggested to be linked to stick-and-slip motility[12].

There is limited information on the function of M2AP in gliding motility and host cell invasion, as its function is solely dependent on its association with MIC2. However, previous research has shown that M2AP binds to the last TSR domain (TSR6) of MIC2[9], and the regions on M2AP responsible for TSR6 binding have been mapped by NMR titrations[13]. Nonetheless, the detailed mechanism of association remains unclear. To gain insights into how M2AP and MIC2 associate with each other, we solved the complex structure of TSR6–M2AP at high resolution. The complex structure explains specializations on TSR6 that are essential for M2AP-binding. We also revealed a remarkable flipped-out conformation for the Tyr-602 of TSR6 in the complex with M2AP, and investigated the dynamics of Tyr-602 by computational modelling and NMR spectroscopy.

## Results

**Crystal structures**. The TSR6 of MIC2 and the core β-domain of M2AP were expressed individually in SHuffle *E. coli* cells and then mixed together in a 1:1 ratio before undergoing gel filtration (Methods). Crystallization trials were performed on the complex as well as on each individual protein. The optimized crystals of M2AP β-domain alone (we referred as M2AP hereinafter) and its complex with TSR6 diffracted to high resolution (1.8–2 Å). However, structure of M2AP alone could not be solved using molecular replacement with the previous NMR model[13]. Thus, experimental phasing with a selenium derivative was used instead (Fig. 1b and Table 1). The asymmetric unit contained three M2AP molecules, which were very similar to each other (Cα r.m.s.d. 0.23–0.39 Å) but less similar to the NMR structure (Cα r.m.s.d. 2.5–3 Å) (Fig. 1c). The crystal and previous NMR structure showed that M2AP has a galectin-like fold comprising 13 antiparallel strands. These strands form two β-sheets that make up two opposite faces, with one face being more hydrophobic than the other (Fig. 1d, e).

The TSR6–M2AP complex structure was solved using the M2AP crystal structure as a search template, and the TSR6 domain was manually built (Supplementary Fig. 1). The final model had $R_{work}$ and $R_{free}$ values of 0.18 and 0.21, respectively (Table 1). Like other TSR domains, TSR6 in the complex has three parallel strands, with a stack of aromatic and basic residues forming π-cation bonds on the layered side, although the number of stacked residues in the layer is lesser in TSR6 than in typical TSR domains (Fig. 2a and Supplementary Fig. 2a). Layered residues are typically conserved among various TSR-containing

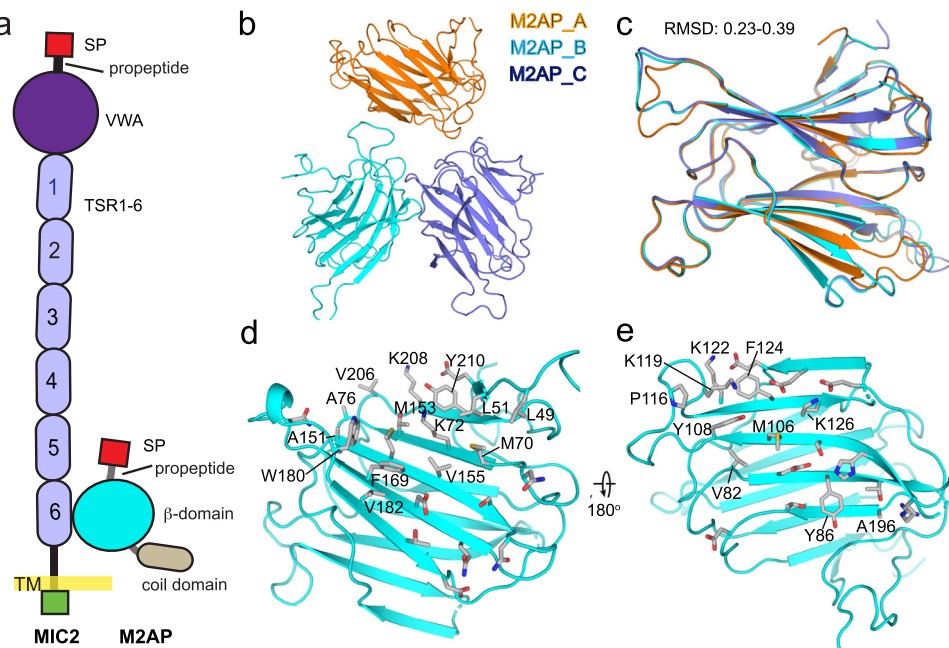

**Fig. 1 Crystal structure of M2AP. a** Schematic diagram of the MIC2-M2AP complex. **b** Overall structure of M2AP alone. **c** Superposition of the three molecules within the asymmetric unit. **d** The more hydrophobic face of M2AP. **e** The less hydrophobic face of M2AP. Surface residues are shown as sticks, with hydrophobic residues are labelled.

**Table 1 Data collection and refinement statistics.**

|  | M2AP | | MIC2$^{TSR6}$-M2AP |
| --- | --- | --- | --- |
| *Data collection* | Se derivative | Native | |
| Space group | C222$_1$ | C222$_1$ | P2$_1$2$_1$2$_1$ |
| Cell dimensions | | | |
| *a, b, c* (Å) | 82.64, 184.15, 84.36 | 82.58, 183.39, 84.93 | 42.78, 53.18, 95.55 |
| α, β, γ (°) | 90, 90, 90 | 90, 90, 90 | 90, 90, 90 |
| Resolution (Å)$^a$ | 92.08–2.34 (2.47–2.34) | 45.85–1.81 (1.91–1.81) | 23.03–2.0 (2.05–2.0) |
| $R_{sym}$ or $R_{merge}$$^b$ | 0.13 (1.429) | 0.053 (1.136) | 0.057 (0.084) |
| CC1/2$^c$ | 0.998/0.783 | 1.00 (0.698) | 0.998 (0.993) |
| I/σ(I) | 14.1 (2.3) | 17.2 (1.5) | 27.2 (16.2) |
| Completeness (%) | 99.7 (99.9) | 99.9 (99.5) | 95.6 (76.1) |
| Redundancy | 13.2 (13.8) | 6.8 (7.0) | 8.8 (6.3) |
| *Refinement* | | | |
| No. reflections | | 59,225 | 14,604 |
| Resolution (Å) | | 45.85–1.81 | 23.03–2.00 |
| $R_{work}$/$R_{free}$ | | 0.180/0.215 | 0.175/0.212 |
| No. of atoms | | | |
| Protein | | 4022 | 1673 |
| Water | | 427 | 270 |
| B factor, all atoms (Å$^2$) | | 46.0 | 16.0 |
| R.m.s deviations | | | |
| Bond lengths (Å) | | 0.011 | 0.002 |
| Bond angles (°) | | 1.195 | 0.519 |
| Ramachandran (%)$^d$ | | 97.7/2.3/0 | 97.2/2.8/0 |

$^a$Values for highest resolution shells are given in parentheses.
$^b$Rmerge = $\sum hkl \sum_i |I_i(hkl) - \langle I(hkl) \rangle| / \sum hkl \sum_i I_i(hkl)$ where $I_i(hkl)$ and $\langle I(hkl) \rangle$ are the i and mean measurement of intensity of reflection $hkl$.
$^c$CC1/2 = Pearson's correlation coefficient between average intensities of random half data sets for each unique reflection.
$^d$Residues in favored, accepted, and outlier regions of the Ramachandran plot as reported by MOLPROBITY.

proteins, whereas residues on the non-layered side are quite diverse. Notably, a presumably layered residue, Tyr-602, adopts a conformation out of the layer, and a water molecule is positioned within the layer and hydrogen bonds to TSR6 residues Thr-601 and Pro-635 (Fig. 2a). The layered side is more electrostatic than the non-layered side (Fig. 2b). The complex structure reveals that the rod-like TSR6 domain binds through its non-layered side onto the hydrophobic face of the M2AP β-domain (Fig. 2c). The binding on the M2AP side is quite rigid, as only slight variations were identified when comparing the complex structure with the M2AP-only structure. Notable differences include the Trp-180 bearing loop region that moves slightly toward the TSR6 (Fig. 2d), consistent with previous literature identifying Trp-180 and its nearby hydrophobic patch as the key region for binding with MIC2[13].

The TSR6–M2AP interface buries a total of 1460 Å$^2$ of accessible surface area. At the center of the binding interface, Lys-72 of M2AP forms a hydrogen bond interaction with both His-620 and the main-chains of Ala-621 and Thr-636 (Fig. 2e). The His-620 of TSR6 additionally forms extensive hydrophobic interactions with M2AP residues Met-70, Met-153, Val-155, and Phe-169. At one edge of the interface, Trp-180 of M2AP inserts into a valley contributed by TSR6 ribbon residues Tyr-602 through Ser-605 and hydrogen bonds to the main-chain of Val-603 (Fig. 2e). At the other edge of the interface, Phe-637 of TSR6 packs against M2AP residues Leu-49, Leu-51, and Met-70 (Fig. 2f). Additionally, Val-623 of TSR6 forms van der Waals interactions with Val-206 and Lys-208 of M2AP, and its main-chain forms a hydrogen bond network mediated by two waters with Thr-74 and Tyr-210 of M2AP. Waters are also involved in the interactions between Thr-636 of TSR6 and Lys-72 of M2AP,

and between His-620 of TSR6 and M2AP residues Lys-72, Tyr-210, Ser-157, and Asp-167 (Fig. 2e, f).

**Mutagenesis validation of the binding interface.** The importance of specific residues in the interface was confirmed mutationally with isothermal titration calorimetry (ITC) (Fig. 3). Wild-type (WT) TSR6 and a control mutation to alanine of a residue outside the interface, Tyr-616 (Y616A), bound M2AP with affinities ($K_D$) of 10–12 nM (Fig. 3a, b). In agreement with the structural findings, S605, V623A, T636A, F637A mutations decreased affinity by ~2, ~3, ~10, and ~4-fold, respectively (Fig. 3c–f). Remarkably, the H620A mutant decreased binding by >5000-fold, showing that His-620 is a hotspot in the binding interface (Fig. 3g). Moreover, combining H620A with a mutation of Thr-636 to Lys (T636K) completely abolished binding activity (Fig. 3h). His-620 is not present in the first 5 TSR domains of MIC2, or other structurally characterized TSR domains (Fig. 4), and thus contributes to specific binding of M2AP to TSR6 of MIC2.

Interestingly, although the structure shows that Tyr-602 of TSR6 points toward Ala-76 and Asn-173 of M2AP and hydrogen bonds to Asn-173 (Fig. 2e), the Y602A and Y602F mutants bind to M2AP with similar affinity as WT TSR6 (Fig. 3i–k). In other TSR domains, aromatic residues in the position of Tyr-602 are generally present in the layer, although in TSR6 the basic residue that would normally stack between Tyr-602 and Trp-604 is also missing (Fig. 2a). To address this puzzle, we used AlphaFold Multimer to predict the structure of TSR6 alone or in complex with M2AP[14]. With TSR6 alone, Tyr-602 located to the layered side in four out of the top five predicted models, between Trp-604 and the C600-C632 disulfide bond, while one model showed Tyr-602 flipped to the other side and in a conformation similar to the structure of TSR6 in the complex (Fig. 4a, left). In contrast, in four out of the top five models in the predicted complex structures, Tyr-602 adopted the flipped-out state, in contrast to only one model showing it on the layered side (Fig. 4a, right). This suggests that Tyr-602 might be able to equilibrate between these two states: in isolation, it favors the layered state, while in the presence of M2AP, Tyr-602 favors the flipped-out state in which its side chain can interact with Lys-624 of TSR6 and some hydrophobic residues (Ala-151, Leu-171, Trp-180) of M2AP.

**The dynamic feature and specializations of TSR6.** $^{19}$F nuclear magnetic resonance (NMR) spectroscopy has been widely applied to investigate protein structure and dynamic changes[15,16]. In order to further investigate the conformational dynamics of residue Tyr-602 on TSR6, we substituted it with the structurally similar L-4-trifluoromethylphenylalanine (tfmF) residue through unnatural amino acid incorporation. The resulting protein was named TSR6-Y602tfmF. TSR6-Y602tfmF $^{19}$F solution NMR spectra were acquired with and without M2AP at room temperature at pH 7.5. The $^{19}$F-NMR spectra of TSR6-Y602tfmF showed two peaks at −61.03 and −61.50 ppm (Fig. 4b). The up-field peak located at −61.50 ppm was assigned to the layered state due to the close contact with W604[17], while the down-field peak at −61.03 was assigned to the flipped-out state.

When M2AP was added to TSR6-Y602tfmF, the flipped-out state was shifted further down-field (−60.37 ppm) and the population of this component increased significantly compared to the population observed in the spectrum of TSR6-Y602tfmF alone (Fig. 4b). This suggests that the flipped-out state of tfmF602 was stabilized by the addition of M2AP, consistent with our crystal structure and modelling. The 0.65 ppm down-field chemical shift in the complex therefore can be attributed to the contacts with M2AP, which alter the local environment around

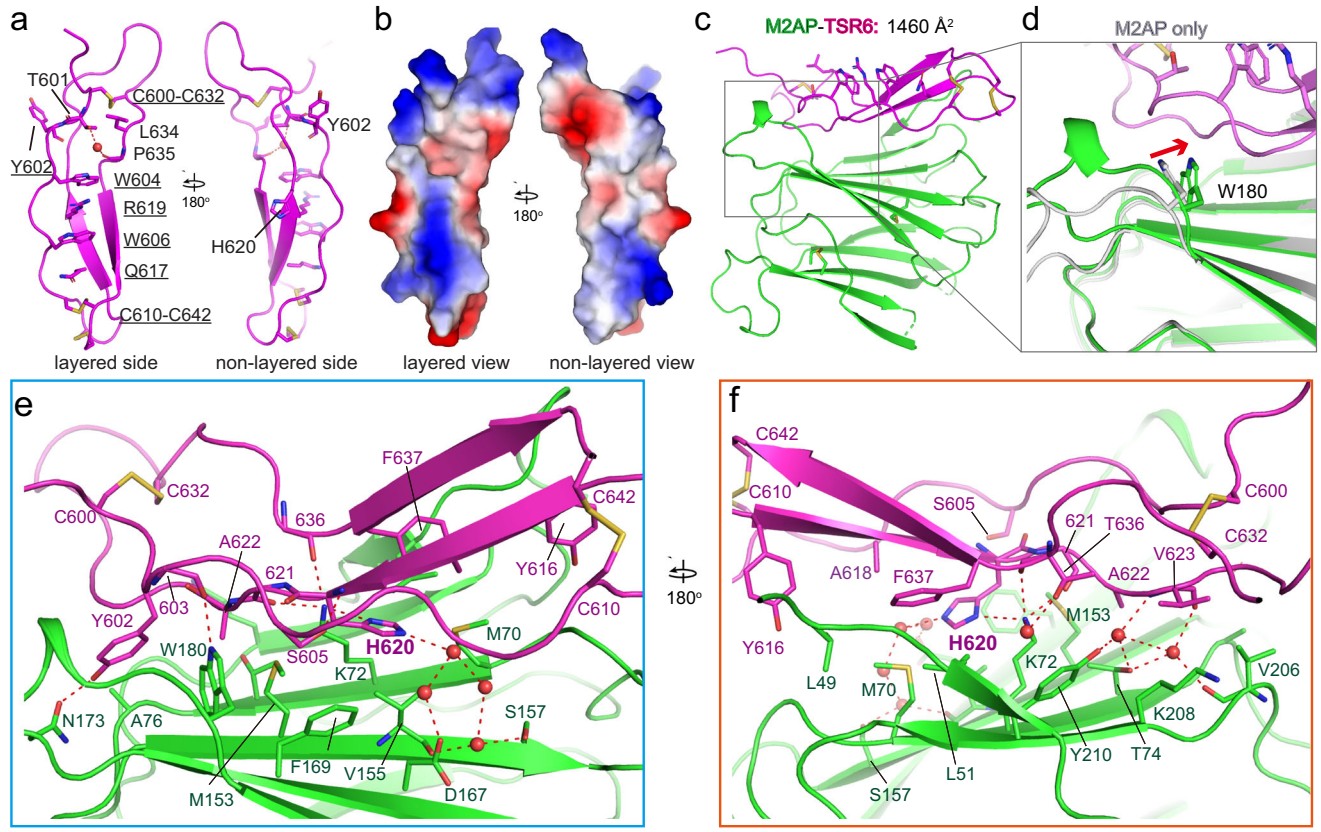

**Fig. 2 The TSR6 structure and binding interface with M2AP. a** TSR6 overall structure. The layered residues are shown as stick and underlined. A water molecule involved in the stack is shown as sphere. **b** The electrostatic surface calculation of TSR6 with scale of [−58 (red) to +58 (blue) kT/e]. **c** Overall binding mode of TSR6–M2AP. **d** The M2AP only crystal (colored gray) was superimposed onto the complex structure for comparison. Arrow indicates significant movement. **e, f** The detailed interactions between TSR6 and M2AP. Interacting residues are shown as sticks, waters are shown as spheres. Red dash lines indicate hydrogen bonds.

tfmF602 compared to the *apo* TSR6 states. Furthermore, there is an attenuated peak at −61.50 ppm in the spectra of the complex, which can be assigned to a small population of layered state for either free TSR6 or in complex with M2AP, because the layered state of tfmF602 shares identical environment with or without M2AP. These NMR findings support the notion that Tyr-602 undergoes conformational equilibration between two states, and that binding to M2AP shifts the equilibrium towards the flipped-out state in which the Tyr-602 participates in the interface. Overall, the NMR data align well with the crystal structure and models, providing further insights into the conformational dynamics of Tyr-602 in TSR6.

The difference in sequence between TSR6 and other representative TSRs may explain the mechanism for dynamic feature of Tyr-602 (Fig. 4c, d). Unlike the TSR domain of TRAP[12] or TSR1 of MIC2[9], which have 7 layered residues, the TSR6 of MIC2 appears to contain 8 layered residues when comparing structure and sequence. This is similar to its homologs in FSP[18] or TSP1[19] (Fig. 4c). However, a key layered Arg/Lys residue is not conserved in TSR6. Instead, it is replaced by a small residue (Ala-621), which creates a 2-layer space between the C600-C632 disulfide bond and the Trp-604 residue. This structural feature generates a metastable region, which makes it easier for Tyr-602 to flip back and forth.

## Discussion

Our MIC2-M2AP complex structure shows how TSR6 specifically binds to M2AP. A key residue for association, His-620, is unique

to TSR6 among MIC2 TSR domains. Histidine in this position is conserved in MIC2 TSR6 orthologs of *Besnoitia* and *Cystoisospora* but is an asparagine in *Neospora* and glutamic acid in *Eimeria* (Supplementary Fig. 3a). Nonetheless, both asparagine and glutamic acid may form similar hydrogen bond interactions with Lys/Arg residues in their cognate M2APs in the same position as *Toxoplasma* M2AP Lys-72. Many other residues within the MIC2-M2AP interface are quite hydrophobic, including Ala-622, Val-623, and Phe-637 of TSR6, and a cluster of nonpolar residues in the binding face of M2AP (Supplementary Fig. 3b). These hydrophobic residues in TSR6 are not conserved in the other 5 MIC2 TSR domains. In fact, TSR6 is more hydrophobic on its non-layered ligand-binding surface compared to other MIC2 TSR domains, as reflected by the calculated grand average hydropathy scores (GRAVY) (Supplementary Fig. 2b). Among the first 5 TSR domains, TSR1 is an exception, as it shows a similar GRAVY score as TSR6. However, the hydrophobicity of TSR1 can be partially complemented by the upstream VWA domain, which covers a specialized hydrophobic pigtail of TSR1 in the MIC2 VWA-TSR1 structure[9]. These differences reveal the specializations of TSR6 for M2AP interaction.

In our previous small angle X-ray scattering (SAXS) experiments, we observed that MIC2 has an elongated conformation with its tail binding to M2AP[9]. Meanwhile, another study also characterized the binding region of M2AP to the TSR6 of MIC2 by NMR titrations[13]. With the newly determined crystal structure of the C-terminal tail region, along with the previously determined N-terminal VWA-TSR1 fragment structure and predicted TSR2-5 models, we have been able to fit these fragments into the

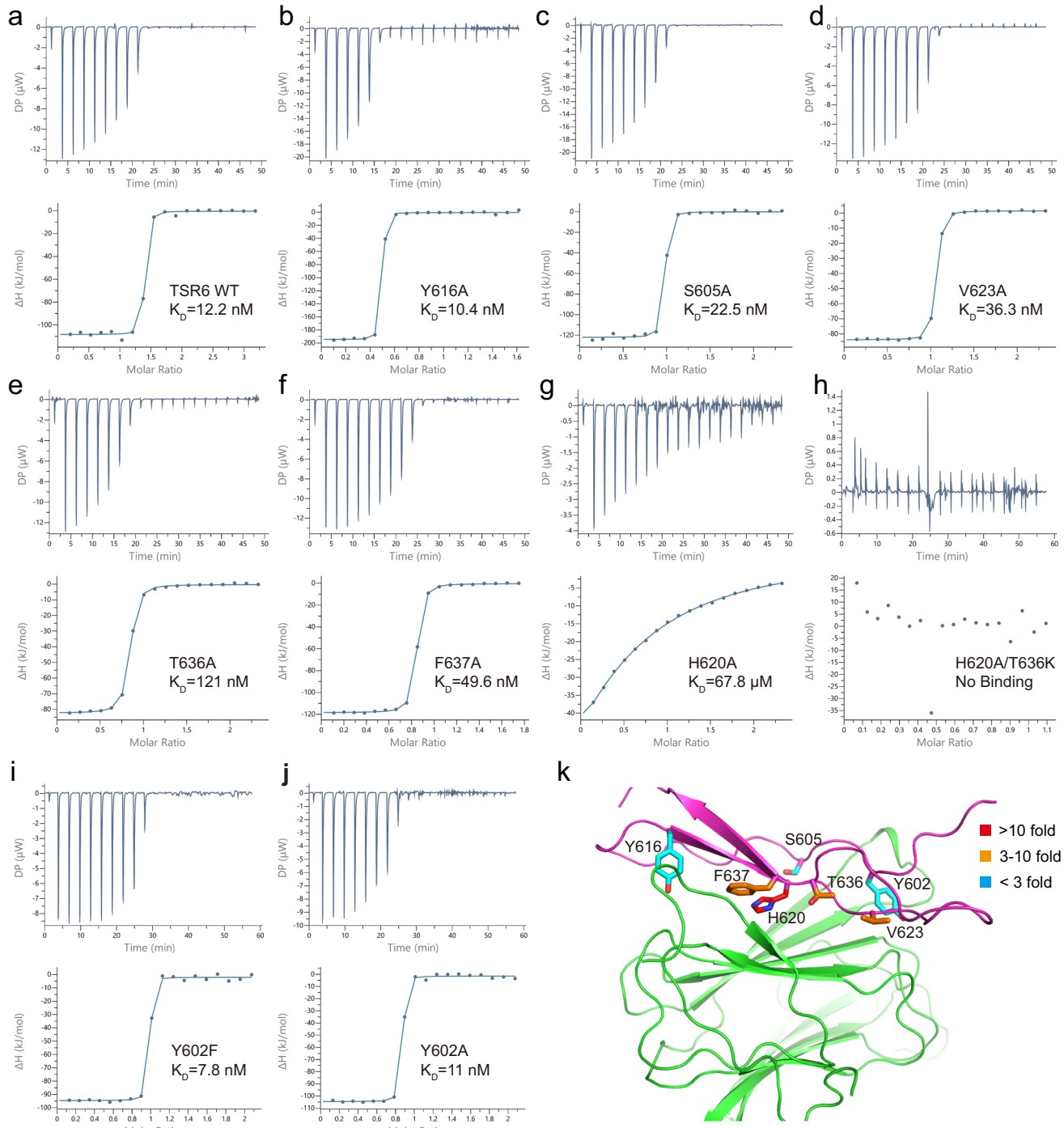

**Fig. 3 Binding profiles of TSR6 mutants with M2AP by ITC. a–j** Binding curves and statistics of M2AP with TSR6 mutations. **k** Map of the tested mutations on the complex structure.

MIC2-M2AP SAXS envelop (Supplementary Fig. 4). The alignment of these fragments suggests some distortion around the TSR3 region, which is consistent with the proline-rich linker sequence between TSR2 and TSR3. We also observed that the tandem TSR domains within MIC2 adopt relatively rigid connections, which may be due to overlaps or interactions between tandem TSR domains. Similar situations have been observed in other tandem adhesion domains, such as EGFs or SCRs[20].

The AlphaFold predicted that the Tyr-602 residue of TSR6 favors the layered state in isolation but flips out when M2AP is associated, this dynamic feature was further validated by NMR studies. The complex models predicted by AlphaFold quite resemble the determined crystal structure, and in 2 out of the top 5 models the His-620 shares exactly identical orientation with that in the crystal structure (Supplementary Fig. 5a–c). Furthermore, we found these models can be successfully used to get the right structural solutions in molecular replacement.

During the conformational change within TSR6, the ~180° flipping of Tyr-602 requires a rotation of its main-chain by a similar degree. Tyr-602 is located in a region (strand 1) in which main-chain rotation is energetically allowed. Unlike other β-strands, the residues on strand 1 of TSR domains primarily donate their side-chains for stack interactions, rather than forming main-chain hydrogen bonds with other β-strands.

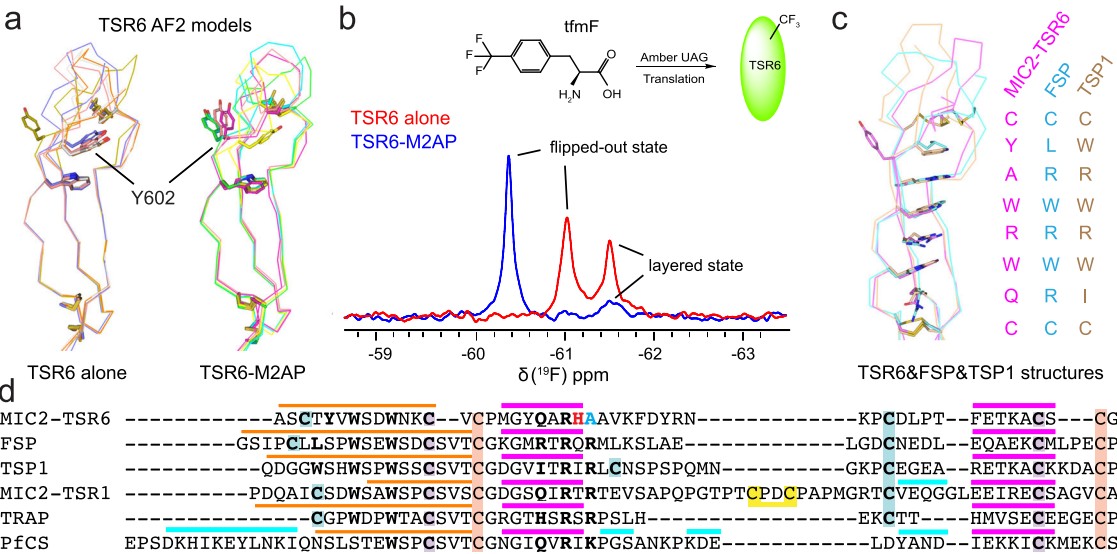

**Fig. 4 The dynamic feature and specializations of TSR6. a** Top five predicted models for TSR6 alone or in complex with M2AP by AlphaFold. **b** $^{19}$F-NMR spectra of TSR6-Y602tfmF alone or in complex with M2AP. Potential layered state and flipped-out state are indicated. Showing on the top is the structure of tfmF and the scheme to incorporate it into synthesized protein. **c** Comparison of the layer of TSR6 with FSP (1VEX) and TSP1 (PDB: 1LSL). **d** sequence alignment of TSR6 with other representative TSR domains. In TSR6, the unconserved residues His-620 (for M2AP-binding) and Ala-621 (for layered residues) are colored red and cyan, respectively, Other layered residues are in bold. Disulfide bonds are backgrounded by paired colors. Above the sequence alignment, orange, pink and cyan lines indicate stand, β-strand, and α-helix, repectively.

Therefore, the flipping of Tyr-602 does not associate with the breakage of its main-chain hydrogen bond network. The unstructured tip region, compared to the well-ordered stranded region, may also facilitate the conformational change around Tyr-602 (Fig. 4). For example, the rotation of Tyr-602 results in not just rotation of Thr-601 but also movement of Leu-634 toward the layer to partially fill the gap left by Tyr-602 (Supplementary Fig. 5d). The Lys-624 also adopts multiple conformations in these predicted models, transiting from a far-most conformation within the Tyr-602 layered states to a position in the flipped-out state forming π-cation interaction with Y602, which also requires ~180° rotation of its side chain (Supplementary Fig. 5d).

We were able to separately express and purify TSR6 and M2AP, despite the predominantly hydrophobic nature of their interface. This apparent contradiction may be reconciled by the presence of two key polar residues, His-620 in TSR6 and Lys-72 in M2AP, located in the middle of the interface, which may facilitate proper behavior of the proteins in isolation. Although the cytoplasmic environment of SHuffle cells does not add glycosylations to target proteins, there is no potential N-glycosylation site within the binding interface, and the O-fucosylation motif, CXX(S/T)C found in many TSR domains, including in Apicomplexans and in MIC2 TSR1-5[9,12], is replaced by a non-fucosylated motif (CVC) in MIC2 TSR6.

Both MIC2 alone and MIC2-M2AP can be secreted successfully in HEK293 cells[9]. These results suggest that although M2AP plays a crucial role in the maturation of MIC2 through the parasite's secretory network[6], it is not necessary for the expression of MIC2 in recombinant expression systems. Hence, there may be other factors besides folding that are required for the transportation of MIC2-M2AP to the microneme. For example, the MIC2 and M2AP each contains a propeptide (Fig. 1a): the M2AP propeptide contributes to efficient trafficking of the TgMIC2-M2AP complex to the micronemes[8], while the MIC2 propeptide was suggested as a trigger for the opening of its N-terminal VWA domain[9]. Furthermore, MIC2 also contains two key motifs in its cytoplasmic domain that are required for targeting to the micronemes[21]. The mutations on M2AP that

break the interaction with MIC2 were reported to retain the complex in nucleus or ER on a parasite level[13], We thus may expect some similar results for TSR6 key residue (like His-620) mutations on MIC2 side.

In summary, our study elucidated the structure of key components in the MIC2-M2AP complex. The crystal structure, along with mutagenesis studies, identified a key determinate residue His-620 in the middle of the interface. Additionally, modelling and NMR studies revealed conformational changes around Tyr-602 in TSR6 during binding with M2AP. The flipping may have functional significance in allowing MIC2 to be stable both in isolation and in complex with M2AP.

## Methods

**Protein expression and purification.** The genes encoding TSR6 (residues 597–651) and M2AP (residues 47–228) were cloned into the modified pMCSG7-His vector and pMGSG7s-His-SUMO vector, respectively. Two mutations (N127R, S177N) were introduced in the M2AP construct to remove potential N-linked glycosylation sites.The plasmids were transformed into *Escherichia coli* strain SHuffle T7 and cultured in LB medium. When the $OD_{600}$ reached 0.6–0.8, isopropyl β-D-1-thiogalactopyranoside was added to a final concentration of 0.4 mM to induce protein expression, and the cells were cultured overnight at 18 °C. The bacterial cells were harvested by centrifugation at 4000 rpm for 50 min at 4 °C. The cell pellet was resuspended in lysis buffer (20 mM Tris-HCl pH 7.5, 500 mM NaCl, 0.1% TritonX-100, 1% Cocktail), and then lysed using a high-pressure homogenizer. Cell debris was immediately removed by centrifugation at 21,000 rpm for 1 h at 4 °C.

The recombinant expressed protein was purified using a Ni-NTA column. The His tag and His-SUMO tag were removed by digestion with Tobacco Etch Virus (TEV) protease and passing through a second Ni-NTA column. Three residues (SNA) from the expression vector was left in the N-terminus of each protein after TEV digestion. The proteins were further purified using a Superdex 75 column (GE Healthcare, United States) which was

pre-equilibrated with buffer (20 mM Tris-HCl pH 7.5 and 150 mM NaCl).

The purified TSR6 and M2AP proteins were mixed in a 1:1 molar ratio at 4 °C overnight and then subjected to a second Superdex 75 column to assemble the protein complex. The Se-Met labeled M2AP protein was expressed in a Se-Met containing M9 medium and purified using the same protocol as the wild-type protein. All mutated variants were generated by a site-directed mutagenesis method and expressed and purified using the same protocol as the wild-type protein.

**Crystallization, data collection, and structure determination**. The purified M2AP, Se-Met labeled M2AP, and TSR6–M2AP complex were concentrated to 12 mg/ml, 4.5 mg/ml, and 11.4 mg/ml, respectively. The concentrated proteins were used for crystallization using hanging-drop vapor-diffusion method with different commercial crystallization screening kits from Hampton Research (State of California, United States). The M2AP was crystallized in a solution containing 0.1 M Hepes sodium pH 6.5 and 1.35 M sodium citrate tribasic dihydrate. Meanwhile, the Se-Met labeled M2AP crystal was generated in the same condition as the wild-type M2AP by sitting-drop vapor-diffusion method. The crystal of TSR6–M2AP complex was obtained in a solution containing 15% glycerol, 0.2 M sodium acetate trihydrate, 0.1 M Tris-HCl pH 8.5, and 30% w/v polyethylene glycol 4000. Crystals were rapidly frozen in liquid nitrogen and all data were collected at beamline BL18U1 of the National Center for Protein Sciences Shanghai (NCPSS) at the Shanghai Synchrotron Radiation Facility (SSRF). The diffraction images were processed, integrated, and scaled using the XDS package[22]. Phase of M2AP was determined using phase information obtained by the anomalous diffraction of selenium measured at wavelength of 0.9791 Å (Se-SAD)[23]. The phase was extended to the high resolution native data and the structural model of M2AP was built in Phenix with autobuild[24], and then manually rebuilt in COOT[25]. The structure of TSR6–M2AP complex was solved by molecular replacement using the solved M2AP structure as a searching model. The TSR6 domain in TSR6–M2AP complex was manually built based on electronic densities. Structure-related figures were generated using Pymol (https://www.pymol.org). The structure-based sequence alignment was conducted using CLUSTALW and presented by ESPript server[26]. The data collection and structure refinement statistics are summarized in Table 1.

**Isothermal titration calorimetry**. The Isothermal Titration Calorimetry (ITC) experiments were performed using a MicroCal PEAQ-ITC instrument (Malvern) at a temperature of 25 °C. The protein samples were dissolved in 20 mM Tris-HCl pH 7.5 and 100 mM NaCl, with TSR6 used as the fixed sample and M2AP as the titration sample. To measure the binding affinity between M2AP and TSR6 WT or mutants, 40 µl M2AP (500–670 µM) in the syringe was titrated into 280 µl TSR6 (30–60 µM) in the cell. Data were analyzed using MicroCal PEAQ-ITC Analysis Software.

**19F-NMR experiments**. The pMCSG7-His-TSR6 with codon for Tyr-602 was mutated to TAG and plasmid pEVOL-tfmF (comprising the coding tRNA$_{CUA}$ and tfmF-specific aminoacyl-tRNA synthetase) were co-transformed into *E. coli* strain SHuffle T7 in the presence of 100 µg/ml ampicillin and 25 µg/ml chloramphenicol[27,28]. The transformed cells were grown at 37 °C in LB medium to an OD$_{600}$ of 0.6, the temperature was reduced to 18 °C, then arabinose and mtfF were added to a final concentration of 0.02% and 1 mM, respectively. When the OD$_{600}$ reached 0.8, a final concentration of 0.02% arabinose and 0.4 mM

IPTG were added to induce the protein expression. Cells were harvested 16 h after induction and the TSR6-Y602mtfF protein was purified as wild-type TSR6 described above.

The fluorine labelled TSR6-Y602tfmF was concentrated to 25 µM TSR6 (500 µl) and the NMR spectra were measured without or with M2AP (molar ratio: 1:1.5). All NMR spectra were recorded on a Bruker AVANCE-600 MHz spectrometer (Bruker Biospin, Billerica, MA) equipped with a TCI $^1$H/$^{19}$F-$^{13}$C-$^{15}$N triple resonance cryoprobe and a shielded z-gradient coil. All samples contained 10 µM TFA, as an internal reference, which was set at -75.425 ppm. 1D $^{19}$F-NMR experiments were recorded at 298 K, with a data size of 8192 complex points and 4096 scans per experiment. All data were processed with the spectra zero-filled to 8192 points, and a 30 Hz exponential line broadening was applied prior to Fourier transformation. The data were analyzed by using Bruker Topspin 4.1.3.

**Statistics and reproducibility**. All ITC and $^{19}$F-NMR experiments were repeated to confirm the reproductivity.

**Reporting summary**. Further information on research design is available in the Nature Portfolio Reporting Summary linked to this article.

## Data availability
Atomic coordinate and structure factor for the M2AP only and in complex with MIC-TSR6 have been deposited in the Protein Data Bank with identification code 8J67 and 8J64, respectively. All data needed to evaluate the conclusions in the paper are present in the paper and/or the Supplementary figures. The source data for Figs. 3 and 4b can be found in Supplementary Data 1 and 2, respectively.

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

## Acknowledgements

We thank the staff of BL17U1 from Shanghai Synchrotron Radiation Source and BL18U1 from the National Center for Protein Sciences Shanghai (NCPSS), for their assistance in data collection and processing. We thank the Instruments Sharing Platform of School of Life Sciences, East China Normal University. This work was supported by the National Natural Science Foundation of China (32171215), the basic research program of Science and Technology Commission of Shanghai Municipality (21JC1402400), and the National Key Research and Development Program of China (2018YFA0507001). We thank Drs. Heng Ru and Suwen Zhao for helpful discussion.

## Author contributions

S.Z. expressed and purified the complex protein for crystallization, conducted mutagenesis and prepared samples for ITC and NMR study; F.W. assisted protein purification, crystal optimization and data collection, conducted ITC experiments and edited manuscript; D.Z. initiated the project and got the crystal structure for M2AP only; D.L. did the NMR experiments and analysed data; W.D. assisted the data collection and structure determination. T.A.S. provided the original plasmids, and edited the manuscript. G.S. supervised the project, refined the crystal structures, and wrote the manuscript. All authors involved in the discussion and provided feedback for the manuscript.

## Competing interests

The authors declare no competing interests.
