## [Peer Review File · Communications Biology]

Reviewers' comments:

Reviewer #1 (Remarks to the Author):

The focus of this paper is the high-resolution crystal structure of the complex between the MIC2-associated protein (M2AP) and the 6th thrombospondin domain (TSR6) of the surface adhesin MIC2 from *Toxoplasma gondii*. This adhesin complex plays a major role in apicomplexan gliding motility. The structure build upon previous work describing the solution structure of M2AP and mapping its interface with TSR6 and provides new atomic insight into this important interaction. A key conclusion is that Y602 adopts a flipped out and a stacked conformation in free TSR6, but in the complex it always flipped out. The functional significance is this switch is attributed to stability of MIC2. While this structural biology study is high quality and should be published, it represents a modest step forward in our structural understanding of the MIC2 presentation on the surface of the parasite. The conclusion of the paper could go a little further with the following suggestion. The same authors have published a SAXS envelope of the full MIC2 ectodomain with M2AP, they could now model this to high resolution using the SAXS data together available crystal structures and AlphaFold models of TSRs where the structures are unknown. This would lift the paper and should be attempted. Furthermore, in this early paper the author tested a longer construct going beyond TSR6 i.e. to 679. Did they test whether this extension contributes to the interaction with M2AP at all?

Reviewer #2 (Remarks to the Author):

The authors in the paper have demonstrated the crystal structure of MIC2-associated protein (M2AP) and M2AP-TSR6 complex. *Toxoplasma gondii* for its gliding motility and successful host cell invasion depends on the microneme proteins. The interplay between Microneme protein 2 (MIC2) and MIC2-associated protein (M2AP) is crucial for the maturation and transport from the microneme to the parasite surface. They have identified His-620 of TSR6 as an important residue that binds to M2AP and have shown that Y602 is dynamic and switch from layered state to flipped-out state when bound to M2AP.

Notes:

1. Authors should provide information about the M2AP protein in the introduction.
2. It would have been nicer if the authors can breakdown the results in different heading in the result section for better understanding.
3. Discussion needs improvement and authors need to discuss and compare with the published paper outcomes.
4. It would have been more impactful if the authors could have shown genetically the effect of the mutations in TSR6 residues on the interaction with the M2AP protein.
5. The authors need to be specific about mentioning the domain names instead of full protein name while referring them for expression and purification, as in line number 67.
6. In the line no. 24, it is M2AP not MA2P. The authors should use the Y-602 or tyrosine-602.

Reviewer #3 (Remarks to the Author):

Review: Structural insights into MIC2 recognition by MIC2-associated protein

Su Zhang et al determine the crystal structure of M2AP alone and in complex with the region of MIC2 for which it has been previously determined to bind. They provide a thorough analysis of these

proteins which play crucial roles in gliding motility and host cell invasion.

Overall, this is a solid structural manuscript and furthers our knowledge of these protein and this interaction. The following are some minor revisions that I would recommend prior to publication of this work:

- There are a number of domains in MIC2, one of which becomes the focus of this work. I feel that, from the text, it is difficult to contextualise this. I would strongly recommend a figure (or adding to some relevant figure) with the domain breakdown of Tg MIC2 (similar to Fig1A in your prior work Song, Springer 2014). As well, a simple schematic contextualising this in the context of its biological function – perhaps as an additional figure in the discussion – would be very helpful in helping readers better understand the importance and physiological relevance of this work.
- In ~line 81 you describe T602 “flip[ing] out of the layer in the complex structure”. This section is discussing the complex structure, and not supported by any structure of MIC2-TSR6 alone (NMR to support this is discussed much later in the manuscript), so it is unclear why this is described as such at this point.
- The labelling for T601 and P635 in panel 2A are also smaller and difficult to read.
- The depiction of W180 in Figure 2C is very small, perhaps a zoom of this region could be a further panel so that this is more easily observed.
- Line 93 – “Other than that...” – this seems a bit informal, I would suggest changing how this sentence starts.
- Out of curiosity, in the discussion you mention that the AF models resemble the complex as determined experimentally, is it possible to phase either or both of your experimental datasets w/ these models? If not, can you speculate why? This information might be interesting to mention in the discussion.
- You speculate at the end that Y602 may be important for ensuring stability in isolation and in complex w/ M2AP. I think this is an interesting point, especially given your results showing that this position tolerates F/A mutations w/o notable impact on overall binding affinities. *I do not think it is required for the sake of publication*, but it might be interesting to assay the thermal stability of the wt vs A vs F substitutions, both in and out of complex – if you have spare protein on hand and access to relevant instrumentation.

Reviewer #1 (Remarks to the Author):

The focus of this paper is the high-resolution crystal structure of the complex between the MIC2-associated protein (M2AP) and the 6th thrombospondin domain (TSR6) of the surface adhesin MIC2 from *Toxoplasma gondii*. This adhesin complex plays a major role in apicomplexan gliding motility. The structure build upon previous work describing the solution structure of M2AP and mapping its interface with TSR6 and provides new atomic insight into this important interaction. A key conclusion is that Y602 adopts a flipped out and a stacked conformation in free TSR6, but in the complex it always flipped out. The functional significance is this switch is attributed to stability of MIC2. While this structural biology study is high quality and should be published, it represents a modest step forward in our structural understanding of the MIC2 presentation on the surface of the parasite.

Response: We thank the reviewer for his/her positive comments.

1. The conclusion of the paper could go a little further with the following suggestion. The same authors have published a SAXS envelope of the full MIC2 ectodomain with M2AP, they could now model this to high resolution using the SAXS data together available crystal structures and AlphaFold models of TSRs where the structures are unknown. This would lift the paper and should be attempted.

Response: We thank the reviewer for his/her very helpful suggestions. We now have added the full-length MIC2-M2AP envelope and fitting model in Figure S4 (also as below Fig. R1) and added the relevant descriptions in the revised manuscript in lines 191-200.

Fig. R1. Full-length MIC2-M2AP envelope and fitting model. A and B. The complex model and SAXS envelop showing extended conformation for MIC2 (Ref. 9). C. Fitting result of previous VWA-TSR1 (ref. 9), TSR6-M2AP (this study) crystal structures, and TSR2-5 models (from AlphaFold) into the SAXS envelop model.

2. Furthermore, in this early paper the author tested a longer construct going beyond TSR6 i.e. to 679. Did they test whether this extension contributes to the interaction with M2AP at all?

Response: We appreciated the reviewer's question. Previous SAXS and Co-IP [ref 9] assays have shown TSR6 is sufficient for interaction with M2AP. In agreement with our research, the TSR6-M2AP interaction was measured to a strong binding affinity of 12nM (Fig. 3A). We previously also did the ITC (below) to test whether the residues

after TSR6 (TSR6 ends at 651) may contribute to the interaction with M2AP. The synthesized polypeptide of P20 (residues 656-675 of MIC2) was titrated into M2AP, the resulting isotherm can not be fitted to a binding model, implying P20 failed to interact with M2AP (Fig. R2). This indicates that TSR6 alone is sufficient to bind M2AP.

Fig.R2. Binding profiles of P20 with M2AP by ITC. The polypeptide of MIC2 (residues 656-675) is synthesized and indicated by P20.

Reviewer #2 (Remarks to the Author):

The authors in the paper have demonstrated the crystal structure of MIC2-associated protein (M2AP) and M2AP-TSR6 complex. *Toxoplasma gondii* for its gliding motility and successful host cell invasion depends on the microneme proteins. The interplay between Microneme protein 2 (MIC2) and MIC2-associated protein (M2AP) is crucial for the maturation and transport from the microneme to the parasite surface. They have identified His-620 of TSR6 as an important residue that binds to M2AP and have shown that Y602 is dynamic and switch from layered state to flipped-out state when bound to M2AP.

Notes:

1. Authors should provide information about the M2AP protein in the introduction.

Response: We have now added a sentence and relevant reference to introduce the schematic information of M2AP in lines 43-44:

“MIC2-associated protein (M2AP) is composed of the secretory leader peptide (SLP), propeptide (Pro), β-sheet domain, and Coil domain [6], which is necessary for MIC2 transport through the secretory network [6-8], and knockout or protein level regulation studies have shown the critical role of the MIC2-M2AP complex [7, 8]”.

2. It would have been nicer if the authors can breakdown the results in different heading in the result section for better understanding.

Response: We thank the reviewer for the valuable suggestion. In the revised version, we have added a few subtitles such as “Crystal structures”, “Mutagenesis validation of the binding interface”, and “The dynamic feature and specializations of TSR6”.

3. Discussion needs improvement and authors need to discuss and compare with the published paper outcomes.

Response: We thank the reviewer for this valuable suggestion. We have revised the discussion and added some new supplement discussion. For example, we tried to fit the new structure and previous N-terminal structure as well as AF models into the SAXS envelop and found good agreement (lines 183-193):

“In our previous small angle X-ray scattering (SAXS) experiments, we observed that MIC2 has an elongated conformation with its tail binding to M2AP [9]. Meanwhile, another study also characterized the binding region of M2AP to the TSR6 of MIC2 by NMR titrations [13]. With the newly determined crystal structure of the C-terminal tail region, along with the previously determined N-terminal VWA-TSR1 fragment structure and predicted TSR2-5 models, we have been able to fit these fragments into the MIC2-M2AP SAXS envelop (Fig. S4). The alignment of these fragments suggests some distortion around the TSR3 region, which is consistent with the proline-rich linker sequence between TSR2 and TSR3. We also observed that the tandem TSR domains within MIC2 adopt relatively rigid connections, which may be due to overlaps or interactions between tandem TSR domains. Similar situations have been observed in other tandem adhesion domains, such as EGFs or SCRs [20].”

Also in the revised manuscript we added lines 224-231: “..For example, the MIC2 and M2AP each contains a propeptide: the M2AP propeptide contributes to efficient trafficking of the TgMIC2-M2AP complex to the micronemes [8], while the MIC2 propeptide was suggested as a trigger for the opening of its N-terminal VWA domain [9]. Furthermore, MIC2 contains two key motifs in its cytoplasmic domain that are required for targeting to the micronemes [21]. The mutations on M2AP that break the interaction with MIC2 were reported to retain the complex in nucleus or ER on a parasite level [13], We thus may expect some similar results for TSR6 key residue (like His-620) mutations on MIC2 side.”

4. It would have been more impactful if the authors could have shown genetically the effect of the mutations in TSR6 residues on the interaction with the M2AP protein.

Response: We appreciated the reviewer’s comment. The complex structure reported in current study showed that the His-620 and surrounded hydrophobic residues in TSR6 are responsible for the binding with M2AP. Also for M2AP, the Lys-72 and its nearby residues including Trp-180 are involved in the interaction, this is in consistent with the previously NMR titration study showing the hydrophobic patch (includes Trp-180) as the key region for binding with MIC2 (ref. 13). Furthermore, in that study they found that the mutations on M2AP (e.g., Trp-180, Phe-169) indeed break the interaction with MIC2 and retain the complex in nucleus or ER on a parasite level. We are regretful since we cannot conduct TSR6 mutagenesis study on a parasite level, but we may expect

some similar results as the mutation results from M2AP. In view of the comment we added a sentence in the discussion section as indicated in last question.

5. The authors need to be specific about mentioning the domain names instead of full protein name while referring them for expression and purification, as in line number 67.
Response: We have redefined the name of domain mentioned in the revised version.

6. In the line no. 24, it is M2AP not MA2P. The authors should use the Y-602 or tyrosine-602.

Response: We are sorry for the mistake. We have corrected this mistake in line 24 now. Meanwhile, to save space people usually use three-letter or single letter abbreviation to represent the corresponding amino acid, so we think current three-letter style is also acceptable.

Reviewer #3 (Remarks to the Author):

Review: Structural insights into MIC2 recognition by MIC2-associated protein

Su Zhang et al determine the crystal structure of M2AP alone and in complex with the region of MIC2 for which it has been previously determined to bind. They provide a thorough analysis of these proteins which play crucial roles in gliding motility and host cell invasion.

Overall, this is a solid structural manuscript and furthers our knowledge of these protein and this interaction. The following are some minor revisions that I would recommend prior to publication of this work:

1. There are a number of domains in MIC2, one of which becomes the focus of this work. I feel that, from the text, it is difficult to contextualize this. I would strongly recommend a figure (or adding to some relevant figure) with the domain breakdown of Tg MIC2 (similar to Fig1A in your prior work Song, Springer 2014). As well, a simple schematic contextualising this in the context of its biological function – perhaps as an additional figure in the discussion – would be very helpful in helping readers better understand the importance and physiological relevance of this work.

Response: We now added a schematic panel for the complex in updated figure 1 (same as below Fig. R3).

Fig. R3. Crystal structure of M2AP.

2. In ~line 81 you describe T602 “flip[ing] out of the layer in the complex structure”. This section is discussing the complex structure, and not supported by any structure of MIC2-TSR6 alone (NMR to support this is discussed much later in the manuscript), so it is unclear why this is described as such at this point.

Response: We appreciated this comment, we have rephrased the sentence to remove the word “flipping” in the revised version: “Notably, a presumably layered residue, Tyr-602, adopts a conformation out of the layer, and a water molecule is positioned within the layer and hydrogen bonds to TSR6 residues Thr-601 and Pro-635 (Fig. 2A).”

3. The labelling for T601 and P635 in panel 2A are also smaller and difficult to read.

Response: We have changed the size of these labellings and updated Fig.2A in the revised manuscript (see below Fig.R4)

4. The depiction of W180 in Figure 2C is very small, perhaps a zoom of this region could be a further panel so that this is more easily observed.

Response: We thank the reviewer for the careful observation. We have revised the figure as suggested in updated figure 2 (Fig. R4)

Figure R4. The TSR6 structure and binding interface with M2AP.

5. Line 93 – “Other than that...” – this seems a bit informal, I would suggest changing how this sentence starts.

Response: We appreciated the reviewer’s comment. The sentence has been rephrased in the revised version and described as “The His-620 of TSR6 additionally forms extensive hydrophobic interactions with M2AP residues Met-70, Met-153, Val-155, and Phe-169.”

6. Out of curiosity, in the discussion you mention that the AF models resemble the

complex as determined experimentally, is it possible to phase either or both of your experimental datasets w/ these models? If not, can you speculate why? This information might be interesting to mention in the discussion.

Response: Upon this suggestion we indeed found the AF2 models can be used as structural templates for MR. The top two models of AF2 complex can successfully used in phaser which showed clear solution. The MR for M2AP only is a little complicate as it contains 3 molecules in an asymmetric unit and two of them are connected with a tNCS, whereas I can still get partial solution in the first test and then get the final solution after a few more tests. Our m2ap only crystal data was harvested two years ago when the AlphaFold was not so popular.

In view of the comment we added a sentence in the discussion section: “The complex models predicted by AlphaFold quite resemble the determined crystal structure, and in 2 out of the top 5 models the His-620 shares exactly identical orientation with that in the crystal structure (Fig. S5B-C). Furthermore, in our preliminary tests, these models can be successfully used to get the right structural solutions in molecular replacement.”

7. You speculate at the end that Y602 may be important for ensuring stability in isolation and in complex w/ M2AP. I think this is an interesting point, especially given your results showing that this position tolerates F/A mutations w/o notable impact on overall binding affinities. *I do not think it is required for the sake of publication*, but it might be interesting to assay the thermal stability of the wt vs A vs F substitutions, both in and out of complex – if you have spare protein on hand and access to relevant instrumentation.

Response: we appreciate the comment. The typical way to do thermal stability assay is using SYPRO orange to label the hydrophobic core during unfolding of protein when temperature is increased. While this method may be perfect for globular proteins but not suitable for TSR6 as it is a flat protein composing three strands and doesn't contain a hydrophobic core. We indeed tried this method but the samples showed bad curves without typical transitions in our tests (Figure R5), hence their T_m were not measurable. This is not the main focus of current study anyhow.

Figure R5. The thermal-stability assay of TSR6 and mutants, and in complex with M2AP.

REVIEWERS' COMMENTS:

Reviewer #1 (Remarks to the Author):

The paper is now acceptable for publication.

Reviewer #2 (Remarks to the Author):

The authors have provided satisfactory responses to the comments and I feel the paper can be accepted in its present form.

Reviewer #3 (Remarks to the Author):

None